# Magnetic Nanoparticles as Mediators for Magnetic Hyperthermia Therapy Applications: A Status Review

Miloš Beković [1,*], Irena Ban [2] , Miha Drofenik [2,3] and Janja Stergar [2,*]

1  Institute of Power Engineering, Faculty of Electrical Engineering and Computer Science, University of Maribor, Koroška cesta 46, 2000 Maribor, Slovenia
2  Laboratory of Inorganic Chemistry, Faculty of Chemistry and Chemical Engineering, University of Maribor, Smetanova ulica 17, 2000 Maribor, Slovenia; irena.ban@um.si (I.B.); miha.drofenik@um.si (M.D.)
3  Jožef Stefan Institute, Department for Materials Synthesis, Jamova 39, 1000 Ljubljana, Slovenia
*  Correspondence: milos.bekovic@um.si (M.B.); janja.stergar@um.si (J.S.); Tel.: +386-2-220-71-72 (M.B.); +386-2-229-44-17 (J.S.)

**Abstract:** This concise review delves into the realm of superparamagnetic nanoparticles, specifically focusing on $Fe_2O_3$, $Mg_{1+x}Fe_{2-2x}Ti_xO_4$, $Ni_{1-x}Cu_x$, and $Cr_xNi_{1-x}$, along with their synthesis methods and applications in magnetic hyperthermia. Remarkable advancements have been made in controlling the size and shape of these nanoparticles, achieved through various synthesis techniques such as coprecipitation, mechanical milling, microemulsion, and sol–gel synthesis. Through this review, our objective is to present the outcomes of diverse synthesis methods, the surface treatment of superparamagnetic nanoparticles, their magnetic properties, and Curie temperature, and elucidate their impact on heating efficiency when subjected to high-frequency magnetic fields.

**Keywords:** magnetic nanoparticles; magnetic hyperthermia; specific absorption rate; calorimetric measurements





## 1. Introduction

Magnetic nanoparticles (MNPs) are nanoscale materials with exceptional properties that find applications across diverse fields, including environmental, biomedical, and clinical domains [1–6]. The size range of MNPs is comparable to that of viruses (20–500 nm), proteins (5–50 nm), or genes (2 nm wide and 10–200 nm long). These nanoparticles possess magnetic characteristics, adhering to Coulomb's law, enabling their manipulation through an external magnetic field. Moreover, their substantial surface area can be effectively utilized for binding diverse biological agents [7–9].

The synthesis of magnetic nanoparticles (MNPs) plays a pivotal role in enhancing their properties and evaluating their potential applications [10–13]. It involves a series of carefully designed processes aimed at tailoring their size, shape, composition, and surface characteristics. In the synthesis of MNPs, the main problem is to control the particle size, which results from the high surface energy of these systems. The spontaneous reduction in surface area in magnetic nanoparticles (MNPs) is driven by interfacial tension, particularly during the initial stages of nucleation, growth, and Ostwald ripening. To ensure the stability of MNPs, it becomes crucial to maintain an appropriate surface area while simultaneously implementing effective protection measures [14–16]. The coating should improve the stability and solubility of MNPs, increase their biocompatibility and target specificity, and prevent agglomeration, oxidation, corrosion, and toxicity [17–23]. The MNPs can be synthesized through many different methods including coprecipitation [24–30], thermal decomposition [31–35], hydrothermal synthesis [36–40], microemulsion [41–44], polyol reduction [45–49], the sol–gel method [50–54], and others [55–63]. The synthesized MNPs are usually coated to ensure a proper surface coating and develop some effective protection

strategies to maintain stability. Depending on the end use of the MNPs, specific surface modification processes are selected [2]. The applied coating strategies can roughly be divided into different groups: synthetic polymer [64,65], natural polymer [27,66], organic surfactants [67,68], inorganic components [69,70], and bioactive molecules and structures [71,72].

Magnetic materials serve as highly effective tools for the magnetic separation of small molecules, biomolecules, and cells. Their unique ability to respond to a magnetic field allows for efficient and precise manipulation in separation processes. Additionally, magnetic materials can be coated with various coatings simultaneously, further expanding their versatility and applicability [73]. In the biomedical field, magnetic particles and magnetic composites are utilized as drug carriers [74–81], as contrast agents for magnetic resonance imaging (MRI) [82–87], and in magnetic hyperthermia (MH) [26,55,77,80,88–98], which is also a focus of this short review.

So far, several methods have been discovered to heat the tissue and destroy tumor cells at the increased temperature. Hyperthermia is an alternative and promising method of treating cancer cells, in which the cancer cells die with minimal damage to healthy cells. Hyperthermia is very popularly used together with radiotherapy and chemotherapy, which is why it has attracted great interest from many researchers from different fields. They started with translational research materials arranged to protect the entire tumor. This method requires carefully planned surgery, but it was not possible to control the temperature.

The basis of MH is the increased heat sensitivity of cancer cells. It has been found that treatment of the cancer area is most effective at a therapeutic temperature between 41 and 46 °C, as it kills most cancer cells in the tissue of interest. This finding was based on the observation that the growth of cancer cells can be stopped at a temperature above 42 °C, while healthy cells survive at higher temperatures. Thermal energy is generated by magnetic particles adhering to diseased tissue exposed to an AMF. The basis of this method is the irradiation of cancerous tissue with implanted magnetic material in the form of magnetic particles or MNPs in an AMF. The amount of thermal energy released depends on the type of magnetic material and the parameters of the magnetic field.

MH is a therapy in which tissue temperature can be increased by exposing MNPs to an alternating magnetic field (AMF). The final heating temperature achieved depends on the Curie temperature ($T_C$) of the MNPs [99]. This method must overcome two problems: the temperature increase must be strictly limited only in the target region so that all other regions are not affected, and the temperature must be controlled inside and outside the target region. Self-regulation MH is a phenomenon in which the $T_C$ changes with the changing chemical composition of the MNPs. Magnetic materials lose their magnetic properties above $T_C$, so magnetic heating is stopped. This type of therapy results in the death of cancer cells with minimal damage to surrounding healthy tissue [100–102].

The most important feature of magnetic hyperthermia is the fact that it is thermal energy generated in a well-defined space associated with the distribution of magnetic particles that we can design. In magnetic hyperthermia, in which magnetic nanoparticles are involved, four different mechanisms are mainly active, depending on the morphology of the magnetic particles and the frequencies of the alternating magnetic field: eddy losses, hysteresis losses, relaxation losses, and resonance losses, but these are small due to the relatively low frequencies. The main characteristic of magnetic particles in an alternating magnetic field is their heating and the conversion of magnetic energy into thermal energy, which heats their surroundings. In particular, the absorption rate of magnetic energy, which is then reflected in the heating of samples, is represented by the parameter "specific absorption rate" (SAR). This describes the heating ability of the corresponding material—MNPs—or the SAR that we consider to be the average absorbed power, not the unit mass of the material at the time it is exposed to a fluctuating magnetic field [103–105].

Despite all the biomedical applications of superparamagnetic nanoparticles, this short review mainly focuses on the superparamagnetic $Fe_2O_3$, $Mg_{1+x}Fe_{2-2x}Ti_xO_4$, $Ni_{1-x}Cu_x$, and $Cr_xNi_{1-x}$ MNPs synthesized by various synthesis methods and their applications in MH.

With this brief review, we have aimed to show the relationship between the synthesis routes, surface chemistry, magnetic properties, and $T_C$, which is near or in the therapeutic range of superparamagnetic nanoparticles. We wanted to explain their influence on the heating efficiency when they are only under the influence of an external alternating magnetic field.

## 2. Syntheses of MNPs

In our study, we employed four distinct synthesis methods to produce four different types of magnetic nanoparticles (MNPs) specifically designed for applications in MH treatments. For the synthesis of maghemite MNPs, we used $FeCl_2 \cdot 4H_2O$ and $FeCl_3 \cdot 6H_2O$. The precursors were prepared by the coprecipitation of $Fe^{2+}$ and $Fe^{3+}$ ions using $NH_4OH$ and heated to 80 °C. Then, the CM-dextran solution was added for over 30 min. The suspension was then stored at 80 °C for 30 min and cooled to room temperature. The suspension was magnetically decanted and washed several times with deionized water [27].

$Mg(NO_3)_2 \cdot 6H_2O$, $Fe(NO_3)_3 \cdot 9H_2O$, and $Ti(C_3H_7O)_4$ were used for the synthesis of the $Mg_{1+x}Fe_{2-2x}Ti_xO_4$ complex ferrite. The precursors were prepared by coprecipitation of $Fe^{3+}$, $Mg^{2+}$, and $Ti^{4+}$ ions in different stoichiometric ratios (x = 0.34; 0.37; 0.40) with a NaOH solution (6 M) and heated to 80 °C. Then, the suspensions were aged at 80 °C for 1 h. The product was washed several times with deionized water and dried [26].

NiCu MNPs were prepared by reducing a Ni, Cu-hydrazine complex using the microemulsion method [41], via reducing a Ni, Cu-oxide mixture in a silica matrix [50,103], obtained with a sol–gel method and by mechanical milling [55]. MNPs with a controlled $T_C$ were prepared by reduction of a Ni, Cu-hydrazine complex synthesized in a compartmentalized state of reverse micelles. We used the phase diagram of water/N-cetyl-N,N,N-trimethylammonium bromide (CTAB), and n-butanol/isooctane as a starting point for the preparation of the microemulsion. We used the titration method to select a suitable composition range that would form a microemulsion in the mentioned system. Microemulsion A was prepared with a 0.3 M solution of $Ni^{2+}$ and $Cu^{2+}$ acidified with HCl and then heated to 60 °C in a water bath. Microemulsion B was then added to microemulsion A, using hydrazine as the metal-containing ligand. This mixture was heated at 60 °C for several hours. After the complex was formed, stirring was continued for another hour and NaOH was added. The color change in the microemulsion to a stable black suspension was indicative of the formation of MNPs. The suspension was then centrifuged and washed several times to sediment the particles [41]. MNPs of NiCu alloy with narrow size distribution were prepared by reduction of the Ni and Cu oxide mixture in a silica matrix obtained by the sol–gel method. $Ni(NO_3)_2 \cdot 6H_2O$, $Cu(NO_3)_2 \cdot 3H_2O$, and citric acid (CA) were dissolved in deionized water. After 15 min, absolute ethanol and tetraethyl orthosilicate (TEOS) were added to the solution with vigorous stirring. The molar ratio of all components, in the case of the Ni:Cu ratio of 67.5:32.5, was Ni:Cu:CA:TEOS:D.I.:ethanol = 0.675:0.325:1.1:2.9:40.6:11.6. The salt was dried at room temperature for 72 h and then calcined at 500 °C for 24 h (air atmosphere). We obtained a powder of nickel and copper oxides in a silica matrix, followed by a reduction in an $Ar/H_2$ atmosphere for one day at 850 °C. We obtained the final product $Ni_{1-x}Cu_x$ MNPs in a $SiO_2$ matrix. By etching, the solution was stirred for 1 day under an Ar atmosphere to remove the $SiO_2$ matrix. The final product was washed several times with etching solution and ethanol and redispersed in ethanol [50]. A series of MNPs from NiCu alloy, which had a $T_C$ in the range of 51 and 63 °C, was prepared in the following article. Using the sol–gel method, we reduced a mixture of nickel and copper oxides in a silica matrix with some minor modifications [106].

$Ni_{1-x}Cu_x$ MNPs (x = 40, 30, 27.5, 27, 25, 20) were prepared by mechanical milling. We milled Cu (grain size < 63 μm) and Ni (grain size < 150 μm) in a SPEX (Metuchen, NJ, USA) 8000M and 8000D at 1425 rpm, using steel vials, while the balls-to-powder ratio was 20:1. We milled for 20 h under an inert $N_2$ atmosphere. We also added NaCl during milling to avoid agglomeration of MNPs [55]. $Cr_xNi_{1-x}$ MNPs were prepared by water-in-oil microemulsions and mechanical milling. The titration method was used to determine the stability range of the microemulsion, with the phase diagram of the composition

of water/CTAB and n-butanol/isooctane serving as a starting point. We prepared two microemulsions, the first was by dissolving aqueous $Ni^{2+}$ (0.4 M) and $Cr^{3+}$ (0.1 M) ions, and the second by adding $NaBH_4$ (0.8 M) to a mixture of CTAB, n-butanol, and isooctane. Both microemulsions were of the same volume; subsequently, after two hours of mixing in an inert $N_2$ atmosphere, a black solution was obtained. The mixtures were centrifuged to separate the black MNPs and washed several times with methanol. Finally, the as-prepared alloy powder was heat-treated at 200, 300, 400, and 600 °C. $Cr_xNi_{1-x}$ MNPs were also prepared by mechanical milling. Cr (particle size < 74 μm) and Ni (particle size < 150 μm) were milled in a SPEX 8000M mill at 1425 rpm, where we used steel vials, while the ratio between balls and powder was 20:1. We milled for 20 h in an inert $N_2$ atmosphere [107].

## 3. Results and Discussion

The synthesized MNPs were characterized by various characteristic methods. We focused on and compared the results of X-ray diffraction analysis (XRD), thermogravimetric analysis (TGA/SDTA), modified thermogravimetric analysis, transmission electron microscopy (TEM), and magnetic and calorimetric measurements.

### 3.1. X-ray Diffraction Analysis

Figures 1–7 show the XRD patterns of as-prepared maghemite and maghemite particles covered with CM-dextran MNPs, (Mg, Ti)-ferrite MNPs, as-prepared NiCu MNPs, NiCu MNPs embedded in $SiO_2$ matrix, and as-prepared NiCr NPs heat-treated. XRD measurements were performed with a D5005 diffractometer (Bruker Siemens) and analyzed with Topas software (Bruker, AXS). All presented MNPs with selected compositions are well crystallized. Crystallite sizes were determined from XRD line broadening using the Sherrer equation (Table 1):

$$d_x = \frac{(0.94 \cdot \lambda)}{(\beta \cdot cos\theta)},$$ (1)

where $d_x$ is crystallite size; $\lambda$ is the wavelength of the radiation; $\beta$ is the line broadening at half the maximum intensity (FWHM) in radians and $\theta$ is the corresponding diffraction angle.

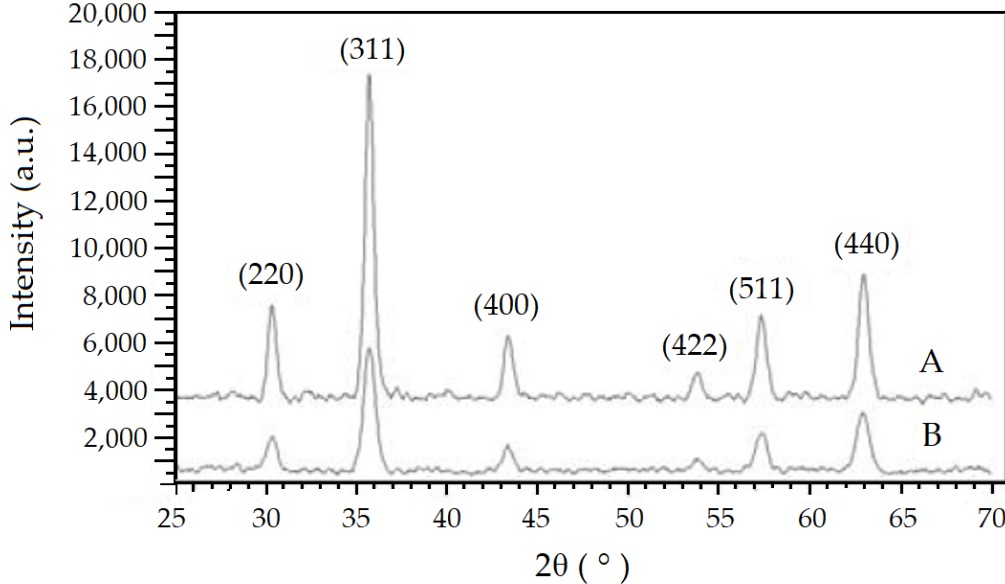

**Figure 1.** XRD spectra of as-prepared maghemite particles (A) and the particles covered with CM-dextran (B).

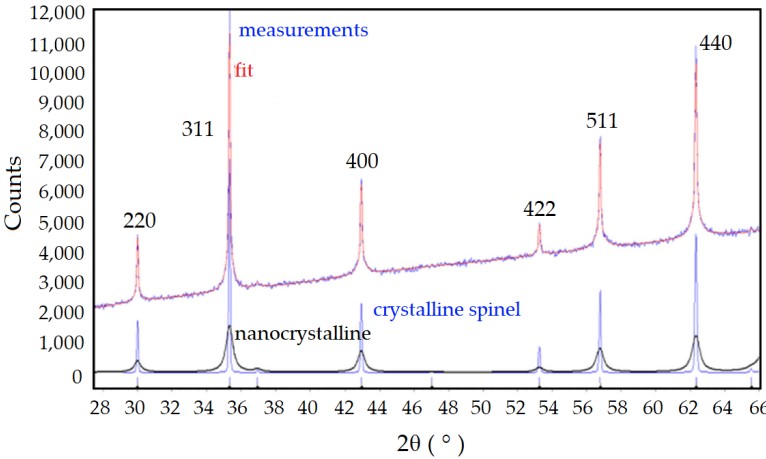

**Figure 2.** XRD spectra of the (Mg, Ti)-ferrite sample (x = 0.37). The curves of the crystalline and nanocrystalline spinel phases are shown below.

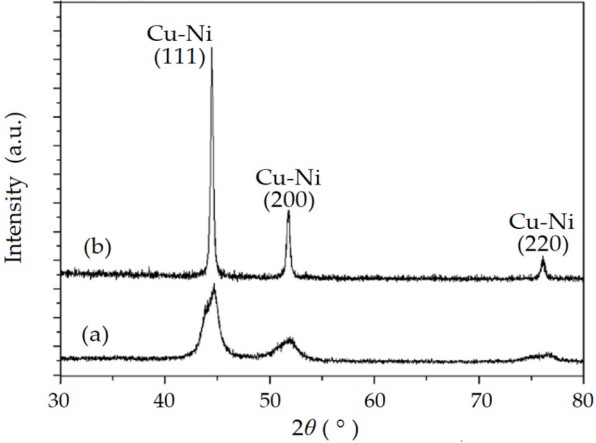

**Figure 3.** XRD spectra of (a) as-prepared $Ni_{0.725}Cu_{0.275}$ MNPs and (b) particles thermally homogenized in NaCl matrix (microemulsion method).

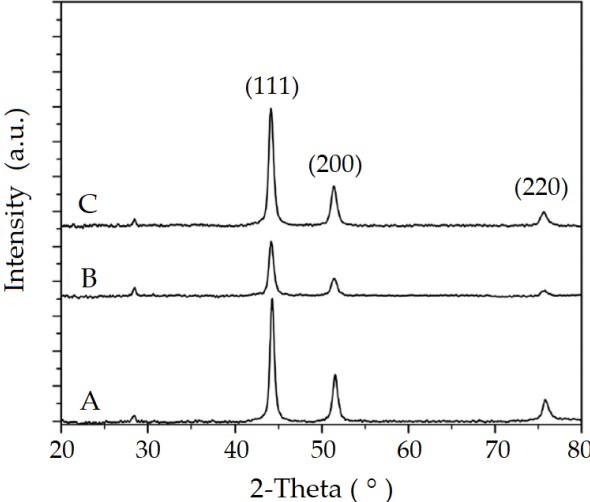

**Figure 4.** XRD powder diffraction patterns of $Ni_{1-x}Cu_x$ ($Ni_{67.5}Cu_{32.5}$ (A), $Ni_{62.5}Cu_{37.5}$ (B), $Ni_{60}Cu_{40}$ (C)) alloy MNPs embedded in the silica matrix (sol–gel method).

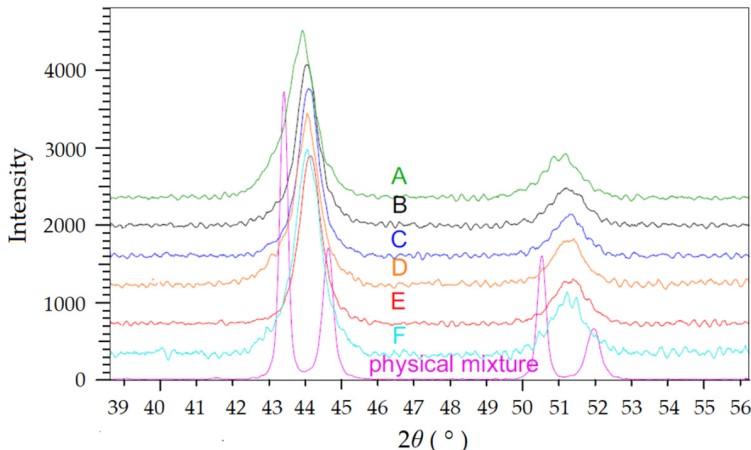

**Figure 5.** XRD patterns of $Cu_{1-x}Ni_x$ samples (milled 20 h) with various compositions. The patterns are compared with the pattern of a Cu–Ni physical mixture, where samples are: A = $Cu_{40}Ni_{60}$, B = $Cu_{30}Ni_{70}$, C = $Cu_{27.5}Ni_{72.5}$, D = $Cu_{27}Ni_{73}$, E = $Cu_{25}Ni_{75}$, F = $Cu_{20}Ni_{80}$.

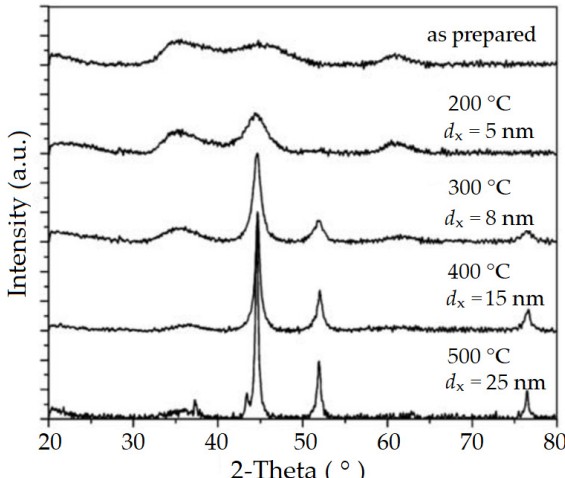

**Figure 6.** XRD patterns of synthesized $Cr_{20}Ni_{80}$ alloy, as-prepared and at different temperatures (microemulsion technique).

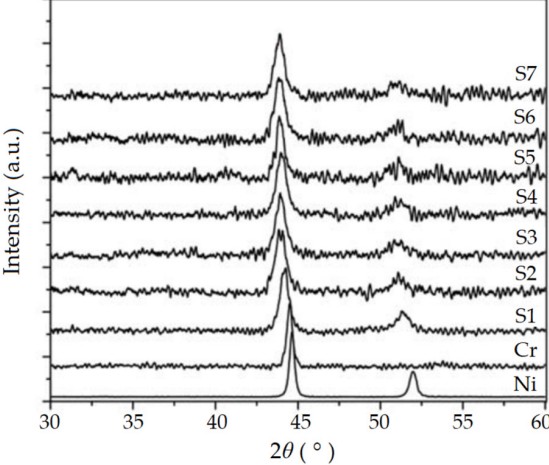

**Figure 7.** XRD patterns of the various $Cr_xNi_{1-x}$ samples were obtained after 20 h of ball milling under an $N_2$ atmosphere. Samples are $Cr_{10}Ni_{90}$ (S1), $Cr_{15}Ni_{85}$ (S2), $Cr_{20}Ni_{80}$ (S3), $Cr_{26}Ni_{74}$ (S4), $Cr_{27}Ni_{73}$ (S5), $Cr_{28}Ni_{72}$ (S6), $Cr_{29}Ni_{71}$ (S7).

**Table 1.** Average particle diameters of different samples were obtained using the Sherrer equation. [108].

| Sample Name | Figure | $d_{\text{XRD}}$ [nm] | References |
|---|---|---|---|
| As prepared maghemite particles (A) | Figure 1 | 11.8 | [27] |
| Maghemite MNPs covered with CM-dextran (B) | Figure 1 | 16.0 | |
| (Mg, Ti)-ferrite MNPs (nanocrystallites) | Figure 2 | 20.0 | [26] |
| (Mg, Ti)-ferrite MNPs (crystalline spinel) | Figure 2 | 200 | |
| As-prepared $Ni_{0.725}Cu_{0.275}$ MNPs | Figure 3 | 7.0 | [41] |
| $Ni_{0.725}Cu_{0.275}$ MNPs thermally homogenized in NaCl matrix | Figure 3 | 28.0 | |
| $Ni_{67.5}Cu_{32.5}$ (A) | Figure 4 | 19.0 | [50,103] |
| $Ni_{62.5}Cu_{37.5}$ (B) | Figure 4 | 17.0 | |
| $Ni_{60}Cu_{40}$ (C) | Figure 4 | 17.0 | |
| $Cu_xNi_{1-x}$ (A–F) | Figure 5 | 10.0–12.0 | [55] |
| $Cr_{20}Ni_{80}$ | Figure 6 | 5.0–25.0 | [107] |
| $Cr_xNi_{1-x}$ (S1–S7) | Figure 7 | 12.0–18.0 | |

The X-ray diffraction patterns of the as-prepared maghemite MNPs (A) and the particles coated with CM-dextran (B) are shown in Figure 1. The diffraction peaks correspond to the cubic phase of maghemite (JCPDS No. 39-1346). Figure 1 clearly shows that the particles are crystalline. Using the Sherrer equation, we estimated the size of the maghemite nanoparticles to be around 16 nm (sample A) and the size of the nanoparticles coated with CM-dextran to be around 11.8 nm (sample B). The size of the nanoparticles is consistent with the analysis of TEM, but one of the main reasons that the coated particles are smaller than the uncoated ones is due to centrifugation, which removed the larger particles [27].

The X-ray diffraction patterns of milled (Mg, Ti)-ferrite sample (x = 0.37) are shown in Figure 2 [26]. The diffraction patterns are consistent with the peaks characteristic of $MgFe_2O_4$, but the diffraction peaks are slightly shifted. The diffraction peaks of $MgFe_2O_4$ (JPCDS 36-0398) have a lattice constant of $a$ = 8.387 Å, those of the (Mg, Ti)-ferrite sample (x = 0.37) have a larger lattice constant of $a$ = 9.418 Å. The XRD data were fitted with a nanoscale crystalline component in addition to a crystalline component, and we can see that the powder consists of two types of crystallites. The crystalline spinel phase with sharp XRD peaks is shown in blue in Figure 2, while the spinel phase is shown in black. The size of nanoparticles is also different in different phases. The size of the nanoparticles in the crystalline spinel phase is about 200 nm, while the size of the nanocrystallites is about 20 nm. According to the XRD analysis, the size of the individual unit cells also varies.

The 2$\theta$ diffraction angles in Figure 3 correspond to the (111), (200), and (220) planes of NiCu alloy crystallites, single FCC phase [41]. Sample (a) in Figure 3 shows as-prepared $Ni_{0.725}Cu_{0.275}$ MNPs (microemulsion method) and sample (b) in Figure 3 shows the particles thermally homogenized in the NaCl matrix. Figure 3 (a) shows the as-prepared nanoparticles synthesized by the microemulsion method. It can be seen that the peaks are broad and therefore were further homogenized under a reducing atmosphere (Ar/$H_2$). Figure 3 (b) clearly shows crystalline peaks that are extremely narrow. Homogenization in a reducing atmosphere also resulted in the growth of nanoparticles. As-prepared nanoparticles were about 7 nm in size, while the homogenized nanoparticles were about 28 nm in size. Thus, the average size of nanoparticles increased at the expense of elevated temperature (750 °C, 5 h) and reduced atmosphere.

The 2$\theta$ diffraction angles in Figure 4 correspond to the (111), (200), and (220) planes of the $Ni_{1-x}Cu_x$ alloy crystallites after reduction in a silica matrix, single-phase composition [103]. The samples are A ($Ni_{67.5}Cu_{32.5}$), B ($Ni_{62.5}Cu_{37.5}$), and C ($Ni_{60}Cu_{40}$). The silica matrix is manifested by a small peak of cristobalite at 2$\theta$ = 28°. Using sol–gel synthesis, we first achieved the formation of copper and nickel oxides [52], which were then heated to 850 °C. Namely, in the silica matrix, the nanoparticles could not agglomerate but only grew accordingly, and the reducing atmosphere ($H_2$/Ar) ensured the formation of $Ni_{1-x}Cu_x$ alloy nanoparticles. Compared with the microemulsion technique, the sol–gel method has

the advantage that the homogenization of nanoparticles does not cause agglomeration and uncontrolled size of nanoparticles.

Figure 4 clearly shows that the nanoparticles are crystalline and have a size between 17 and 19 nm, which was estimated using the Sherrer equation. Moreover, the analysis of TEM shows that the particles are not agglomerated in the silica matrix using the sol–gel method and subsequent homogenization.

In Figure 5, we see the characteristic peaks corresponding to the (100) and (200) planes of bulk FCC metals, 43.47 °, 50.38 ° (Cu), and 44.6 °, and 51.91 ° (Ni) (PDF files 001-1241 and 001-1260). Using EDS analysis, we confirmed that the composition of Cu-Ni MNPs is consistent with the $[Cu^{2+}]:[Ni^{2+}]$ molar ratio used in the synthesis. Figure 5 shows samples that were milled for 20 h, as this time was found to be optimal for the formation of the CuNi alloy. It is clear from the X-ray diffraction patterns that as the copper content increases, the $d_{111}$ spacing also increases. For samples A through F, we estimated particle size between 10 and 12 nm using Sherrer's equation, and the size was consistent with the size of nanoparticles estimated by magnetic measurements [55].

The XRD patterns in Figure 6 show three characteristic broad peaks at $2\theta = 44.57°$, $51.94°$, and $76.51°$. The as-prepared MNPs were amorphous, and upon heating with an elevated temperature, the size of the crystallites or crystallinity increased [107].

The composition of CrNi MNPs (microemulsion technique) was following the $[Cr^{3+}]:[Ni^{2+}]$ = 20:80 molar ratio used for the synthesis. The microemulsion technique for the synthesis of NiCr nanoparticles also proved not to be the best, as extremely amorphous nanoparticles were obtained without thermal homogenization. At the same time, the XRD results also show a peak at 37°, indicating the formation of NiO despite the inert atmosphere throughout the synthesis process. Only with the help of homogenization at elevated temperatures did we obtain crystalline nanoparticles that grew with increasing temperature. Unfortunately, homogenization leads to agglomeration of nanoparticles and at some locations also to larger grains in the form of platelets, which was confirmed by TEM analysis.

In Figure 7, we see the XRD results showing the formation of $Cr_xNi_{1-x}$ alloy using mechanical milling [107]. With increasing Ni content (in %), a progressive diffraction angle was observed in both cases (111) and (200). NiCr samples prepared using mechanical milling were also milled for 20 h in the same manner as NiCu. Figure 7 shows that the Bragg peaks in both cases (111) and (200) shift to higher diffraction angles with increasing Ni content. The lattice constant and interplanar distance $d_{111}$ also increase with increasing chromium content. In the microemulsion method, particles with a size of up to 25 nm were obtained, while in mechanical milling the particles were micrometer in size, which were comprised of nanocrystallites.

### 3.2. Thermogravimetric Analysis and Modified Thermogravimetric Analysis

Figures 8–13 show the TGA/SDTA analysis and thermomagnetic curves, respectively, and the corresponding $T_C$ of CM-dextran-coated maghemite MNPs, (Mg, Ti)-ferrite MNPs, and thermally homogenized NiCu MNPs and ball-milled NiCu and NiCr MNPs. TGA/SDTA analysis was performed using the TGA/SDTA 851$^e$, Mettler Toledo (Columbus, OH, USA). The thermomagnetic (TM) curves and corresponding $T_C$ were determined by thermal demagnetization using a modified TGA on a TGA/SDTA 851$^e$, Mettler Toledo. $T_C$ was measured by placing a permanent magnet on top of the device.

Figure 8 shows the amount/concentration of CM-dextran absorbed into the maghemite MNPs [27]. During thermal heating from room temperature to 750 °C, there is a total weight loss of 21.2%. The weight loss of 16.4% due to the oxidation of the CM-dextran occurs in the first temperature range (150–500 °C), indicated by an exothermic peak of the simultaneous differential thermal analysis (SDTA). The last step, 500–750 °C, is associated with a weight loss of 3.1%, which is due to the removal of the CM-dextran covalently bound to the surface of the particles.

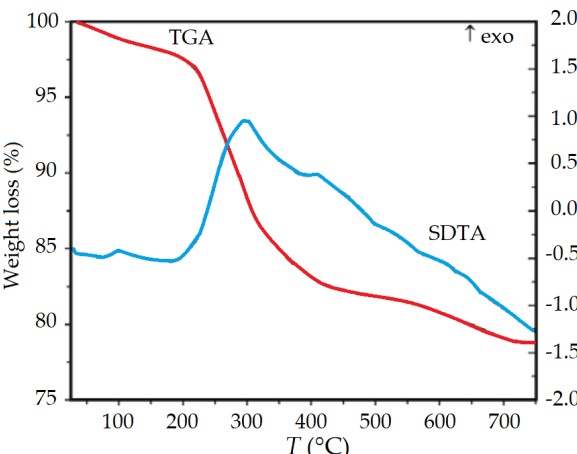

**Figure 8.** Thermal analysis of the maghemite MNPs coated with CM-dextran and the corresponding SDTA.

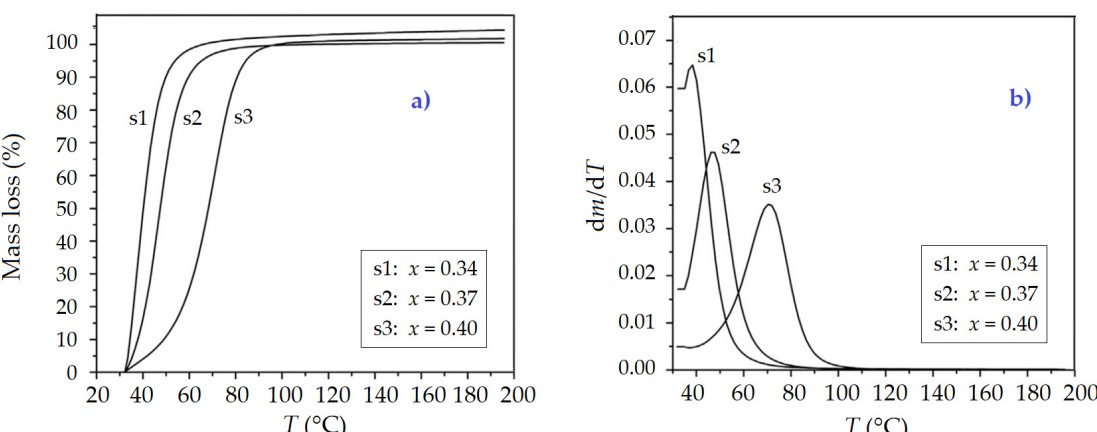

**Figure 9.** (**a**) TM curves of the ferrite particles s1, s2, s3 and (**b**) the corresponding $T_C$ (the first derivative curve).

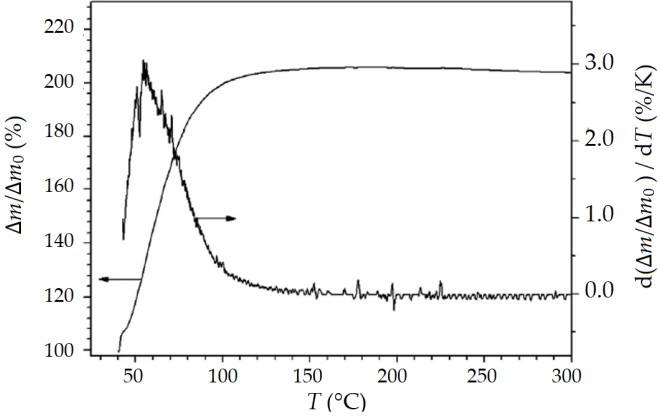

**Figure 10.** TM curve of the thermally homogenized particles $Ni_{0.725}Cu_{0.275}$.

Figure 9 shows the TM curves and the corresponding $T_C$ of as-calcined (Mg, Ti)-ferrite powders (30 min, 1000 °C in the air) [26]. The $T_C$ of MNPs was determined using differential thermomagnetic curves (DTMC), which show an apparent change in weight because of a decrease in magnetization due to an increase in temperature (Figure 9a). Figure 9b shows the first derivative in the TM curve of (Mg, Ti)-ferrite MNPs, the maximum of which is attributed to the corresponding $T_C$. Figure 7 shows the samples for three different compositions. The most promising sample was s2 (x = 0.37), which has a $T_C$ of about 46 °C,

which is ideal for use in MH. This is the temperature at which the nanoparticles transition from a ferromagnetic to a paramagnetic state, meaning that they no longer heat up under the influence of a magnetic field.

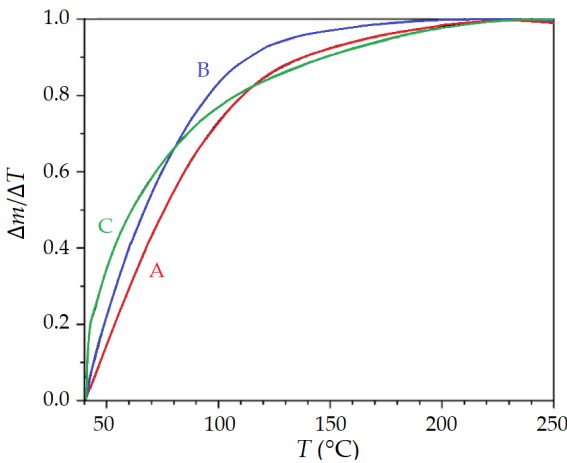

**Figure 11.** TM curves of analyzed samples (A) ($Ni_{67.5}Cu_{32.5}$), (B) ($Ni_{62.5}Cu_{37.5}$), and (C) ($Ni_{60}Cu_{40}$).

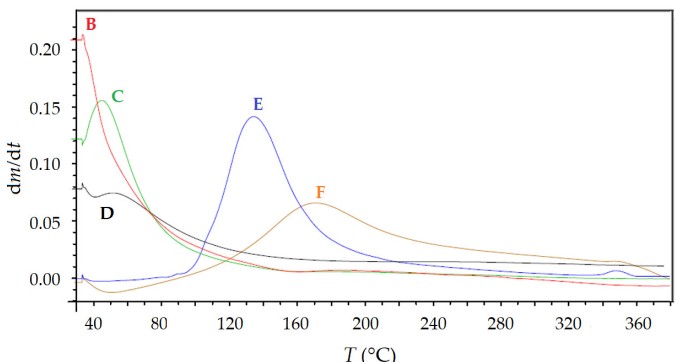

**Figure 12.** First derivative curves of TM curves for $Cu_{1-x}Ni_x$ (B–F) alloys.

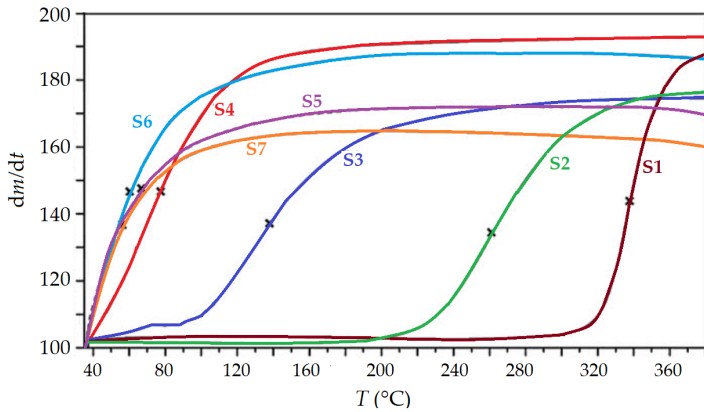

**Figure 13.** TM curves of $Cr_xNi_{1-x}$ samples (S1–S7).

Figure 10 shows the TM curve and its derivative for the thermally homogenized for 5 h at 750 °C under a reducing atmosphere of $Ar/H_2$ alloy $Ni_{0.725}Cu_{0.275}$ MNPs (microemulsion method) [41]. For the as-prepared sample, the $T_C$ was about 250 °C, close to the $T_C$ of Ni, due to the different standard electrode potentials of Cu and Ni or to inhomogeneities in the sample itself. Using homogenization, we then determined the correct $T_C$ for the $Cu_{27.5}Ni_{72.5}$ composition, which was about 45 °C, and which is in the therapeutic temperature range when we refer to MH.

Figure 11 shows the $T_C$ as the temperature at which half of the magnetization weight gain/loss was recorded during heating [103]. Samples A ($Cu_{67.5}Cu_{32.5}$), B ($Ni_{62.5}Cu_{37.5}$), and C ($Ni_{60}Cu40$) were analyzed after heat treatment (850 °C in a reducing atmosphere, 6 h). Nanoparticles synthesized by the sol–gel method have a $T_C$ in the therapeutic range for use in MH. However, homogenization in a reducing atmosphere is required. In Figure 11, the $T_C$ of the samples follows each other, namely sample A has a $T_C$ of 63 °C, sample B has a $T_C$ of 54 °C, and sample C has a $T_C$ of 51 °C. As the Ni content increases, so does the $T_C$.

Figure 12 shows the first derivative of TM curves of $Cu_{1-x}Ni_x$ alloys (ball-milling), the maximum of which is attributed to the corresponding $T_C$ [55]. The TM curves have a slightly asymmetric shape, indicating inhomogeneity in the composition and size of the MNPs. Again, $T_C$ increases with the increasing Ni content, so we can change the composition according to the desired $T_C$. $T_C$ in the therapeutic range around 45 °C shows sample C with composition $Cu_{27.5}Ni_{72.5}$.

Figure 13 shows the DTMC of $Cr_xNi_{1-x}$ alloys (ball-milling) [107]. The $T_C$ of the samples was determined by the temperature, which belongs to 50% of the peaks of the demagnetization curves. The samples are heterogeneous in composition, as evidenced by asymmetric TM curves. Figure 13 also clearly shows how the $T_C$ increases with increasing Ni content in each sample. Sample S1, which has the highest Ni content, also has the highest $T_C$, and conversely, S7 has the lowest $T_C$ and the lowest Ni content.

The $T_C$ for the synthesized (Mg, Ti)-ferrite MNPs, thermally homogenized NiCu MNPs, and NiCu and NiCr ball-milled MNPs were measured and are shown in Table 2.

**Table 2.** $T_C$ of (Mg, Ti)-ferrite and NiCu, NiCr MNPs synthesized with different methods.

| Sample Name | Figure | $T_C$ [°C] | References |
|---|---|---|---|
| (Mg, Ti)-ferrite (x = 0.37) | Figure 9b | 46 | [26] |
| $Ni_{0.725}Cu_{0.275}$ | Figure 10 | 45 | [41] |
| $Ni_{67.5}Cu_{32.5}$ (A) | Figure 11 | 63 | |
| $Ni_{62.5}Cu_{37.5}$ (B) | Figure 11 | 54 | [103] |
| $Ni_{60}Cu_{40}$ (C) | Figure 11 | 51 | |
| $Cu_{30}Ni_{70}$ (B) | Figure 12 | 24 | |
| $Cu_{27.5}Ni_{72.5}$ (C) | Figure 12 | 45 | |
| $Cu_{27}Ni_{73}$ (D) | Figure 12 | 53 | [55] |
| $Cu_{25}Ni_{75}$ (E) | Figure 12 | 137 | |
| $Cu_{20}Ni_{80}$ (F) | Figure 12 | 174 | |
| $Cr_{10}Ni_{90}$ (S1) | Figure 13 | 340 | |
| $Cr_{15}Ni_{85}$ (S2) | Figure 13 | 262 | |
| $Cr_{20}Ni_{80}$ (S3) | Figure 13 | 138 | |
| $Cr_{26}Ni_{74}$ (S4) | Figure 13 | 69 | [107] |
| $Cr_{27}Ni_{73}$ (S5) | Figure 13 | 52 | |
| $Cr_{28}Ni_{72}$ (S6) | Figure 13 | 44 | |
| $Cr_{29}Ni_{71}$ (S7) | Figure 13 | 43 | |

The (Mg, Ti)-ferrite MNPs were synthesized by the coprecipitation method, thermally homogenized $Ni_{0.725}Cu_{0.275}$ were synthesized by a microemulsion method, and $Ni_{67.5}Cu_{32.5}$, $Ni_{62.5}Cu_{37.5}$, and $Ni_{60}Cu_{40}$ in a silica matrix were synthesized by a sol–gel method. The NiCu (B–F) (Figure 12) and NiCr (S1–S7) (Figure 13) MNPs were synthesized by mechanical milling. The transition from a ferrimagnetic to a paramagnetic state at $T_C$ depends on energy exchange. The (Mg, Ti)-ferrite MNPs (sample B, x = 0.37) exhibit $T_C$ close to 46 °C and thus meet the therapeutic requirement. To obtain the desired $T$c, MNPs with composition $Cu_{27.5}Ni_{72.5}$ (microemulsion technique) were thermally homogenized at 750 °C for 5 h in a reducing $Ar/H_2$ atmosphere. The homogenized MNPs have $T_C$ at 45 °C. For the MNPs ($Ni_{67.5}Cu_{32.5}$ (A), $Ni_{62.5}Cu_{37.5}$ (B), $Ni_{60}Cu_{40}$ (C)) synthesized by the sol–gel method, the determined $T_C$ agreed with the selected nominal composition. In the case of mechanical milling, the measurements showed that the $T_C$ of MNPs could be adjusted by changing the molar ratio of Cu/Ni. $T_C$ increased with increasing nickel content. Similar

results were obtained in the case of mechanical milling of the synthesized CrNi MNPs. For NiCu and NiCr nanoparticles, the table and individual figures clearly show that $T_C$ increases with increasing Ni content. $T_C$ for pure Ni is about 357 °C. $T_C$ has a few samples in the therapeutic range for use in MH, but not all of them are suitable in terms of their shape or dispersion.

*3.3. Transmission Electron Microscopy*

Figures 14–17 show the TEM analysis of as-synthesized maghemite and the CM-dextran-coated maghemite MNPs, (Mg, Ti)-ferrite MNPs, as-prepared and thermally homogenized $Ni_{0.725}Cu_{0.275}$ MNPs, and of NiCu MNPs embedded in the silica matrix. The nanoparticle size and the crystallinity were characterized by TEM using a JEOL 2010F microscope. The MNPs were deposited on a cooper-grid-supported, perforated, transparent carbon film.

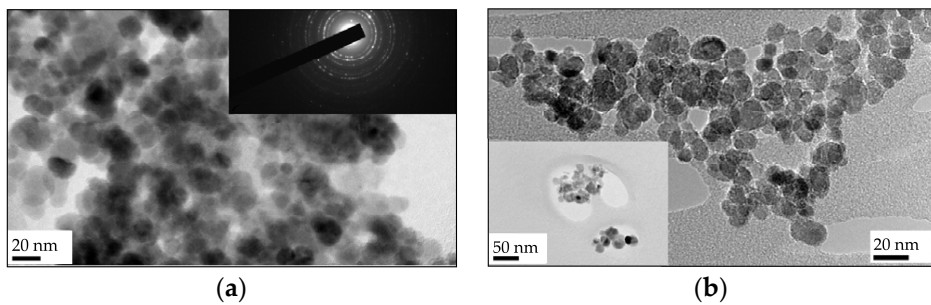

**Figure 14.** TEM micrographs of (**a**) as-prepared maghemite particles (A), (**b**) coated maghemite with CM-dextran.

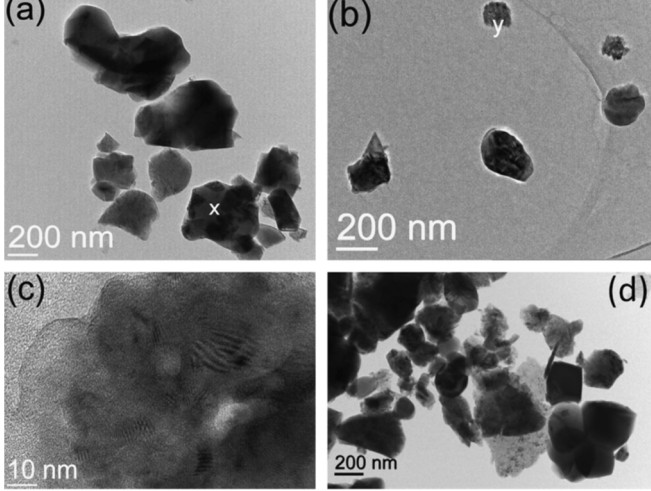

**Figure 15.** Particles of sample B with mechanically strained particle = x (**a**), consisting of nanocrystallites = y (**b**), enlarged view (**c**), and a typical image of the milled powder (**d**).

Figure 14 shows as-prepared maghemite MNPs (a) and the maghemite NP coated with CM-dextran (b) [27]. The main reason that the coated particles have a smaller average particle size is that they were centrifuged, which removes the larger particles from the magnetic fluid. Using TEM images, we also determined the size of the nanoparticles, which was about 14.5 nm for the maghemite nanoparticles and 12.0 nm for the maghemite nanoparticles coated with CM-dextran. When we compare the size estimated with Sherrer's equation, the results are consistent with or very similar to the XRD results.

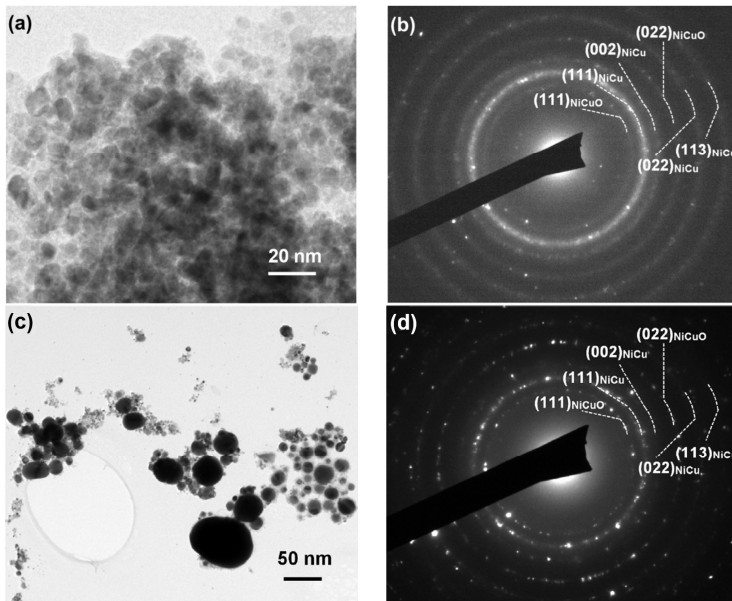

**Figure 16.** TEM image (**a**) and corresponding electron-diffraction pattern of the as-prepared MNPs (**b**) (microemulsion method). TEM image (**c**) and corresponding electron-diffraction pattern of the thermally homogenized particles (**d**).

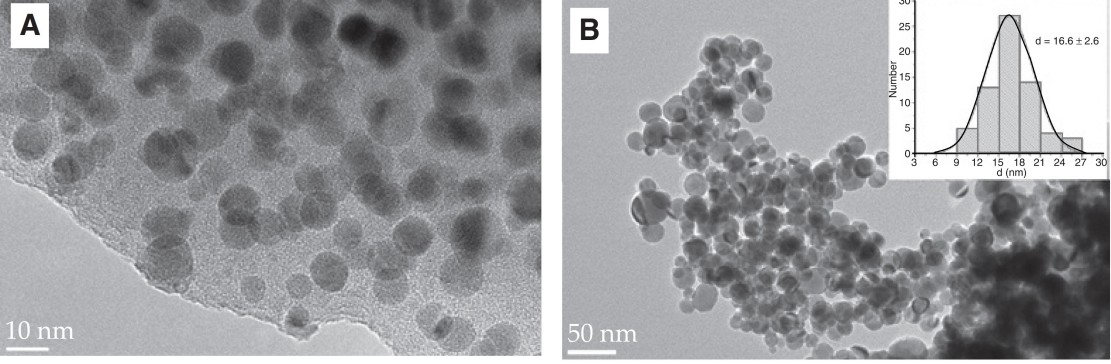

**Figure 17.** (**A**) Typical TEM images of MNPs of sample $Ni_{67.5}Cu_{32.5}$ embedded in a silica matrix and (**B**) bare alloy particles and the corresponding histogram.

Figure 15 shows the (Mg, Ti)-ferrite MNPs with a size of a few hundred nanometers [26]. The particle marked with x (b) shows contours that could have been formed by the mechanical strain. According to what was said or found by XRD analysis, TEM images confirm the formation of crystalline components and nanosized crystalline components. So, we can confirm with certainty that the sample consists of two types of crystallites. In Figure 15a, we see very large nanoparticles larger than a hundred nanometers, while in Figure 15b,c we see nanocrystallites.

In Figure 16 (above), we see a TEM image of prepared NiCu MNPs (microemulsion method) and the corresponding electron diffraction [41]. The corresponding electron diffraction shows a combination of two cubic FCC structures (Ni-Cu, NiCuO). Using TEM images, we were able to determine the size of the nanoparticles, which ranged from 3 to 10 nm. Figure 16 (below) shows a TEM image of thermally homogenized MNPs (750 °C, 5 h under $H_2$/Ar atmosphere) and their corresponding electron diffraction showing larger agglomerated particles of Ni-Cu and NiCuO structure. Homogenization resulted in a relatively broad distribution of nanoparticle sizes, ranging from a few tens of nm to several hundreds of nm. The samples were homogenized, but on the other hand, homogenization led to agglomeration, which we do not want in the case of use in MH.

Figure 17A shows the NiCu alloy MNPs embedded in the silica matrix (sol–gel method) [103]. Spherical, non-agglomerated grains with a relatively uniform size distribution were observed. The sol–gel method proved to be very promising as the nanoparticles were spherical despite homogenization and no agglomeration occurred due to the silica matrix. Figure 17B shows a TEM image of the bared MNPs after leaching. Using a TEM image, we also determined the size of the nanoparticles after leaching (Figure 17B), the nanoparticles are about 16.6 nm in size, which is consistent with the results of XRD analysis, or later, as we will see, with magnetic measurements.

In Figure 18, we see a TEM image of mechanically milled Cu-Ni, sample C ($Cu_{27.5}Ni_{72.5}$) [55]. The size distribution is broad, the MNPs are partially agglomerated, and longer platelets of 200 nm length and 5 nm thickness are visible. During mechanical milling we see that we do not obtain homogeneous nanoparticles, but nanoparticles with different shapes and sizes.

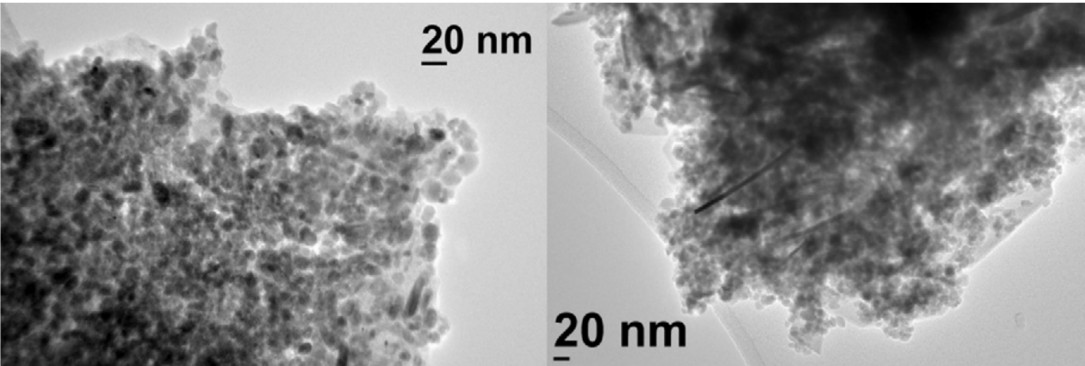

**Figure 18.** TEM image of milled $Cu_{27.5}Ni_{72.5}$ MNPs (sample C, sol–gel method).

The transmission electron micrograph of the $Cr_{20}Ni_{80}$ MNPs synthesized using the microemulsion method and heat treated at 400 °C is shown in Figure 19, left [107]. Like the previous figure, we see a broad size distribution, the particles are agglomerated, and larger grains are observed in some places. In this case of NiCr nanoparticles, mechanical milling also proved to be less than optimal when homogeneity is required, i.e., nanoparticles of the same size and shape.

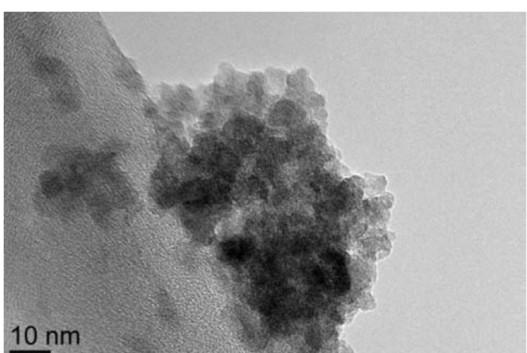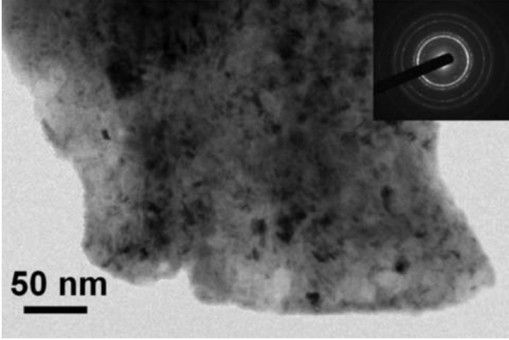

**Figure 19.** TEM micrograph of $Cr_{20}Ni_{80}$ MNPs synthesized using microemulsion method and heat-treated at 400 °C (**left**) and TEM image of mechanically alloyed $Cr_{29}Ni_{71}$ (**right**).

Figure 19, right, shows a TEM image of $Cr_{29}Ni_{71}$ MNPs synthesized by the ball milling method [107]; here, we see micrometer particles with nanocrystallites. The nanocrystallites are between 5 and 30 nm in size.

The particle size was also determined from the TEM analysis (Table 3).

**Table 3.** Average particle diameters of different samples were obtained using TEM.

| Sample Name | Figure | $d_M$ [nm] | References |
|---|---|---|---|
| As-prepared maghemite MNPs | Figure 14a | 14.5 | [27] |
| CM-dextran-coated maghemite MNPs | Figure 14b | 12.0 | |
| (Mg, Ti)-ferrite MNPs (x = 0.37) | Figure 15 | a few hundred nm | [26] |
| $Ni_{0.725}Cu_{0.275}$ | Figure 16a | 3.0–10.0 | [41] |
| $Ni_{0.725}Cu_{0.275}$ | Figure 16c | a few tens to several hundreds of nm | |
| $Ni_{67.5}Cu_{32.5}$ | Figure 17 | 16.6 | [103] |
| $Cu_{27.5}Ni_{72.5}$ (C) | Figure 18 | 10 | [55] |
| $Cr_{20}Ni_{80}$ | Figure 19, left | 5–10 | [107] |
| $Cr_{29}Ni_{71}$ | Figure 19, right | 5–30 | |

Figure 14a,b show typical TEM images of the as-synthesized sample (A) and the CM-dextran-coated maghemite MNPs (B). The average particle diameters estimated from TEM images are 14.5 nm (sample A) and 12.0 nm (sample B), which is in good agreement with the XRD analysis. The (Mg, Ti)-ferrite MNPs are several hundred nanometers in size. The TEM analysis confirmed the XRD results. The MNPs synthesized by the microemulsion method (as-prepared, $Ni_{0.725}Cu_{0.275}$) are about 3 to 10 nm in size. The TEM analysis supports the XRD spectra. The TEM analysis agrees with the XRD analysis. The thermally homogenized MNPs ($Ni_{0.725}Cu_{0.275}$) prepared by the microemulsion method show a relatively broad size distribution (a few tens of nm to several hundred nm). The peaks of the thermally homogenized particles were much sharper, with an estimated value of 28 nm, as indicated by the XRD analysis in agreement with the TEM analysis. The MNPs synthesized by the sol–gel method (after leaching) have a relatively narrow distribution, with an average size of 16.6 nm. The size distribution of $Cu_{27.5}Ni_{72.5}$ (C) MNPs synthesized by mechanical milling is comparable to the XRD analysis and the Sherrer equation, and the average value is 10 nm. The MNPs size distribution of CrNi MNPs (heat-treated at 400 °C) synthesized by the microemulsion technique is relatively broad, with an average particle size comparable to XRD analysis and Sherrer's equation, between 5 and 10 nm. The nanocrystallites sizes of the MNPs ($Cr_{29}Ni_{71}$) synthesized by mechanical milling determined from TEM range from about 5 nm to 30 nm.

*3.4. Magnetic Measurements*

Figures 20–24 show the magnetic properties of maghemite MNPs and maghemite MNPs coated with CM-dextran, (Mg, Ti)-ferrite MNPs, as-prepared and thermally homogenized NiCu MNPs (microemulsion technique), NiCu MNPs (sol–gel method), and ball-milled NiCu MNPs. The magnetic properties of the MNPs were investigated using a Lake Shore 7307 vibrating sample magnetometer.

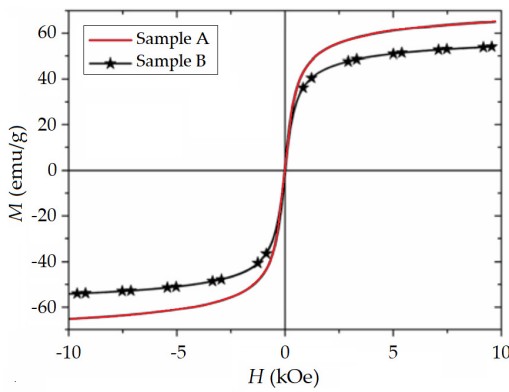

**Figure 20.** Magnetization of as-prepared maghemite MNPs (Sample A) and CM-dextran-coated MNPs (Sample B) at 300 K.

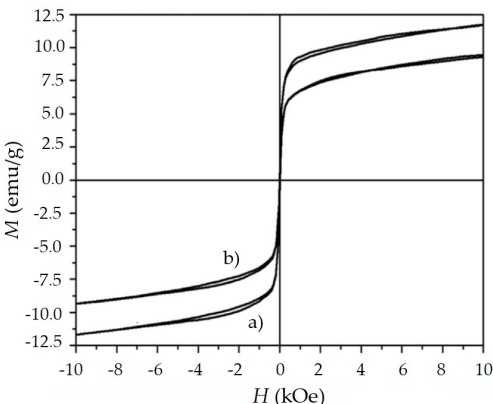

**Figure 21.** Hysteresis for (Mg, Ti)-ferrite as-synthesized (a) and milled (b) powder.

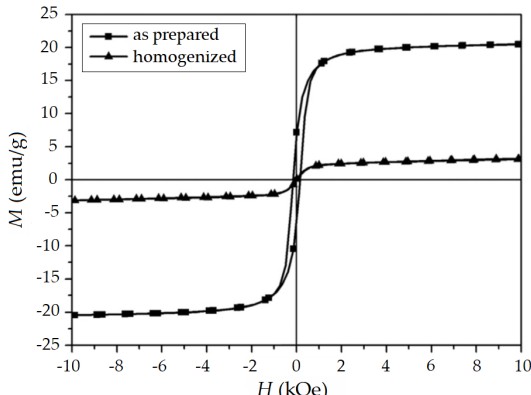

**Figure 22.** Magnetization vs. the AMF for the as-prepared and homogenized NiCu MNPs (microemulsion method).

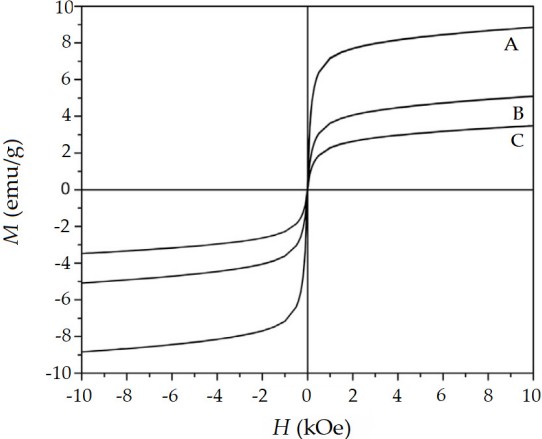

**Figure 23.** Magnetization of synthesized MNPs (sol–gel method) at 293 K: $Ni_{67.5}Cu_{32.5}$ (A), $Ni_{62.5}Cu_{37.5}$ (B), and $Ni_{60}u_{40}$ (C) vs. the magnetic field.

Figure 20 shows the magnetization vs. the magnetic field of the as-synthesized and CM-dextran-covered maghemite particles [27]. The uncoated particles show no hysteresis and have a saturation magnetization ($M_s$) of 65.2 emu/g. The magnetization of the coated MNPs was about 55 emu/g, which is lower than that of the uncoated sample. The average particle diameter of our particles determined by magnetic measurements was 8.20 nm (sample A) and 8.17 nm (sample B). Comparing the size with the XRD and the TEM analysis, certain sizes of nanoparticles are slightly smaller in the magnetic measurements, because the so-called "dead layer" is not considered in the magnetic measurements.

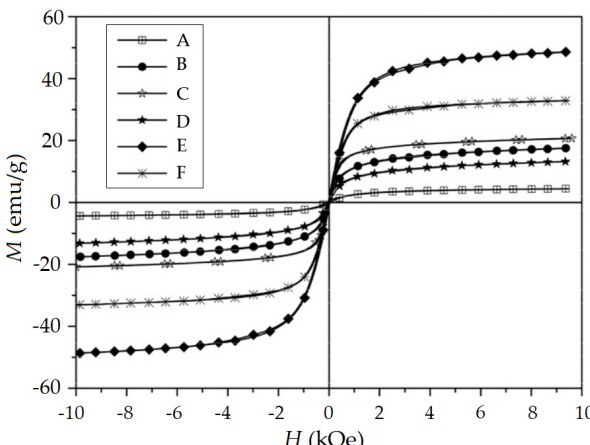

**Figure 24.** Magnetization vs. magnetic field of $Cu_{1-x}Ni_x$ alloy particles, samples (A–F) ball-milled for 20 h.

The magnetization vs. magnetic field curve for the (Mg, Ti)-ferrite particles measured at room temperature under an applied magnetic field of 1 T is shown in Figure 21 [26]. The magnetization of the as-prepared samples at 1 T is higher than the magnetization of the milled sample, which is 8 emu/g. This reduction in magnetization due to milling is a well-known phenomenon. It is caused by induced mechanical stresses and an increase in the specific surface area, consequently, an increase in the non-magnetic fraction ("dead layer"), which is usually located on the surface of the particles and is the result of imperfect coordination and, finally, a non-collinear spin configuration.

Figure 22 shows the magnetic measurements indicating that the magnetization is greatly reduced in the stabilized samples [41]. The MNPs in the as-prepared state exhibit $M_s$~20 emu/g, while it decreases to ~2.5 emu/g in the stabilized MNPs. To obtain homogenized nanoparticles that have a $T_C$ in the therapeutic range, homogenization is necessary, which of course contributes to the drop in magnetization in homogenized nanoparticles.

Figure 23 shows the magnetization of the bare A, B, and C MNPs synthesized by a sol–gel method [103]. The magnetization was measured concerning the magnetic field at room temperature. It can be seen from the figure that the magnetizations measured at $H = 10$ kOe are good approximations for $M_s$ since the slopes of the curves are low under the field $H = 6$ kOe. The considered samples have a similar size, which shows that the size does not influence the magnetization as much as the influence of the different compositions in the $Ni_{1-x}Cu_x$ alloy. The average particle size determined by magnetic measurements was 15.4 nm ($Ni_{67.5}Cu_{32.5}$ (A)), 16.4 nm ($Ni_{62.5}Cu_{37.5}$ (B)), and 17.6 nm ($Ni_{60}Cu_{40}$ (C)).

Figure 24 shows the magnetization vs. the magnetic field of the $Cu_{1-x}Ni_x$ alloy particles [55]. The measurements were performed at room temperature and under the influence of an AMF of 10 kOe. The hysteresis loop is not expressed in any sample, indicating the superparamagnetic properties of the $Cu_{1-x}Ni_x$ MNPs prepared by mechanical milling. The $M_s$ of the samples ranges from 2 to 50 emu/g and depends on the nickel concentration of each sample.

### 3.5. Calorimetric Measurements

Figures 25–27 show the calorimetric measurements of maghemite MNPs coated with CM-dextran, (Mg, Ti)-ferrite MNPs, NiCu MNPs synthesized by microemulsion technique and sol–gel method, and ball-milled NiCu and NiCr MNPs. Magnetic heating effect measurements were performed in a conventionally built calorimetric system that generates an AMF with a nominal field strength of 2 kA/m up to 42 kA/m in the frequency range from 80 kHz to 800 kHz and equipped with a fiber optic temperature probe to measure the temperature required to determine the specific absorption rate (SAR).

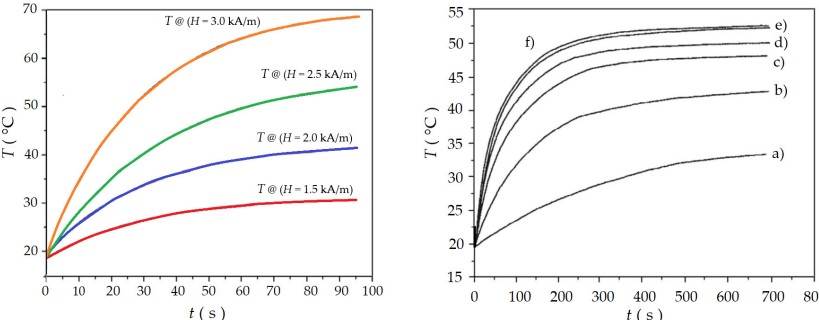

**Figure 25.** Time dependence of the heating of maghemite nanoparticles coated with CM-dextran (sample B) under the influence of different AMF intensities—(**left graph**)—and the calorimetric measurements of a sample (Mg, Ti)-ferrite (x = 0.37, sample B), at different magnitudes of AFM—(**right graph**) ((a) 8.4 kA/m; (b) 12,7 kA/m; (c) 16.9 kA/m; (d) 21.7 kA/m; (e) 25.4 kA/m; (f) 29.6 kA/m).

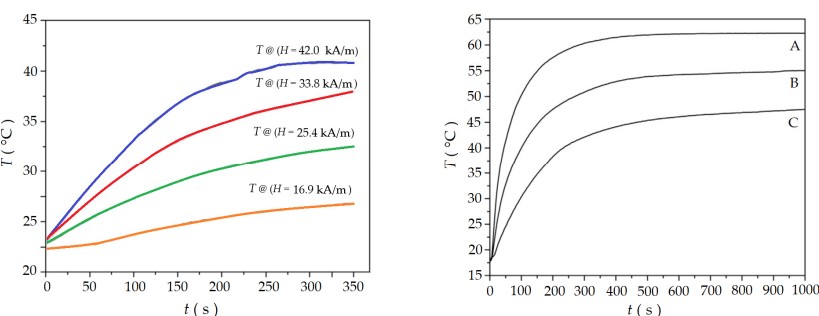

**Figure 26.** Calorimetric measurements for the $Ni_{0.725}Cu_{0.275}$ sample were performed under the influence of different AMF amplitudes—(**left graph**)—and the calorimetric measurements of samples A ($Ni_{67.5}Cu_{32.5}$), B ($Ni_{62.5}Cu_{37.5}$), and C ($Ni_{60}Cu_{40}$) at a magnetic field of 29.4 kA/m and a frequency of 100 kHz—(**right graph**).

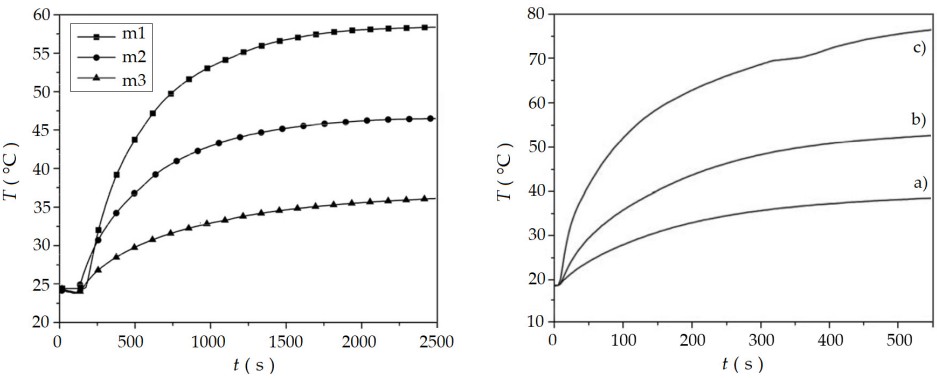

**Figure 27.** Time function of self-heating temperature under the influence of a magnetic field of varying strength for $Cu_{27.5}Ni_{72.5}$ (sample C) at AMF m1- *H* = 3.83 kA/m; m2- *H* = 3.13 kA/m; m3- *H* = 2.41 kA/m—(**left graph**)—and time function of self-heating temperature under the influence of a magnetic field of varying strength for $Cr_{28}Ni_{72}$ at AMF (a) *H* = 9.20 kA/m; (b) *H* = 12.52 kA/m; (c) *H* = 16.18 kA/m—(**right graph**).

In Figure 25, left graph, a time dependence of the heating of maghemite nanoparticles coated with CM-dextran (sample B) under the influence of different AMF intensities at 104 kHz [27] can be seen. The values of SAR were calculated based on the calorimetric response at a frequency of 104 kHz and in the range of 1.5 to 3 kA/m.

The SAR was estimated as:

$$\text{SAR} = \left( \frac{C_{\text{solvent}}}{x_{\text{iron oxide}}} \right) \left( \frac{\Delta T}{\Delta t} \right), \tag{2}$$

where $c_{\text{solvent}}$ is the heat capacity of the solvent ($c = 4.18$ J/Kg for water); $x_{\text{iron oxide}}$ is the weight fraction of iron oxide; and ($\Delta T/\Delta t$) is the initial slope of the time dependence of the self-heating temperature. After a certain period of self-heating, the steady state was achieved when generated heat at a constant temperature was equal to that lost to the environment due to thermal convection. The maximum temperature of this steady state is important when iron oxide-based magnetic fluids (MFs) with a high $T_C$ are used for in vivo applications, as this is the only way to use such an MF without damaging the surrounding tissue during MH.

Figure 25, right graph, shows tested MNPs and the corresponding MF for the SAR [26]. From the course, the SAR was estimated as:

$$\text{SAR} = \left( \frac{C_{solvent}}{x_{(Mg,\,Ti)-ferrite}} \right) \left( \frac{\Delta T}{\Delta t} \right), \tag{3}$$

where $c_{solvent}$ equals $c_{\text{water}}$ 4.18 J/kg; $x_{(Mg, Ti)\text{-ferrite}}$ is the weight fraction of ($Mg, Ti$)-ferrite; and ($\Delta T/\Delta t$) is the initial heating slope as a function of time. SAR values were estimated between 1.2 and 3.9 W/g (8.4 kA and 29.6 kA/m). The result of calorimetric measurements shows that the stationary temperature is higher than the planned $T_C$ (sample B, $x = 0.37$).

In Figure 26, left graph, a calorimetric measurements for the $Ni_{0.725}Cu_{0.275}$ sample performed under the influence of AMF with an amplitude between 16.9 and 42.0 kA/m and a frequency of 104 kHz [41] can be seen. The SAR was estimated as:

$$\text{SAR} = (c_{\text{Ni}}) \left( \frac{\Delta T}{\Delta t} \right), \tag{4}$$

where $c_{\text{Ni}}$ represents the heat capacity of nickel and is 0.444 J/kg. $\Delta T/\Delta t$ represents the initial slope of the time dependence of the self-regulating temperature. After a certain time, the curve reaches a maximum corresponding to the temperature, $T_C$, already determined by thermomagnetic measurements. Table 4 lists the corresponding SAR values.

**Table 4.** SAR values as a function of AMF amplitude for $Ni_{0.725}Cu_{0.275}$ (microemulsion method).

| $H$ [kA/m] | SAR [mW/g] |
|---|---|
| 16.9 | 4.3 |
| 25.4 | 21.8 |
| 33.8 | 35.8 |
| 42.0 | 41.6 |

In Figure 26, right graph [103], we determined the initial heating slopes ($\Delta T/\Delta t$) that we used to determine the values of SAR:

$$\text{SAR} = \left( c_{\text{alloy}} \right) \left( \frac{\Delta T}{\Delta t} \right) \left( \frac{1}{x_{\text{Ni}}} \right), \tag{5}$$

where $x_{\text{Ni}}$ is 0.67, the calculation was based on the mass fraction of nickel in each sample. Using the heat capacities of $c_{\text{Ni}}$ (0.444 J/gK) and $c_{\text{Cu}}$ (0.390 J/gK), we calculate the heat capacity of the alloy, at a temperature of 20 °C, since all measurements had started at this value, defined by the temperature of the surrounding thermal bath.

Table 5 lists the corresponding SAR values.

**Table 5.** Samples of composition $Ni_{1-x}Cu_x$ (sol–gel method) and data of SAR measurements.

| Composition | SAR [W/g] |
|---|---|
| $Ni_{67.5}Cu_{32.5}$ (sample A) | 0.60 |
| $Ni_{62.5}Cu_{37.5}$ (sample B) | 0.36 |
| $Ni_{60}Cu_{40}$ (sample C) | 0.12 |

To conduct calorimetric measurements, a solid powder sample was subjected to controlled heating. These measurements took place within a system capable of generating a nominal field strength of 2 kA/m and operating within a frequency range of 80 to 800 kHz. This setup allowed for precise characterization of the thermal properties and behavior of the sample under varying magnetic field conditions. Figure 27, left graph, shows the growth of the temperature of the powder sample in the calorimeter for different magnetic fields [55]. Remarkably, the $Cu_{27.5}Ni_{72.5}$ MNPs exhibited a significant heating effect during our experiments. Under an applied magnetic field strength of 3.13 kA/m, we were able to achieve temperatures surpassing 40 °C.

Figure 27, right graph, shows the temperature rise of the $Cr_{28}Ni_{72}$ sample in the calorimeter at different magnetic field strengths [107]. The magnetic measurements of the solid powdered $Cr_{28}Ni_{72}$ with a $T_C$ of 44 °C were performed in a system that generates an AMF (9.2–16.2 kA/m, 800 kHz).

## 4. Conclusions

The potential of superparamagnetic nanoparticles in magnetic hyperthermia holds great promise. In this concise review, we explored different synthesis methods employed for the preparation of $Fe_2O_3$, $Mg_{1+x}Fe_{2-2x}Ti_xO_4$, $Ni_{1-x}Cu_x$, $Cr_xNi_{1-x}$ MNPs, their resulting size, shape, functionalization, $T_C$, magnetic, and calorimetric properties. The results indicate that the described synthesis method of superparamagnetic nanoparticles with adjustable Curie temperature offers a convenient approach for producing controlled $T_C$ nanoparticles. These synthesized particles demonstrate excellent suitability, particularly when optimized for MH applications.

Through the coprecipitation method, maghemite nanoparticles were successfully synthesized, and a supplementary CM-dextran coating was applied. The resulting nanoparticles displayed a distinct crystalline structure, as confirmed by both XRD and TEM analysis. The magnetization ranged from 55 to 65 emu/g, with the results of calorimetric measurements showing that the size of the nanoparticles and the thickness of the CM-dextran coating affect the SAR.

The (Mg, Ti)-ferrite nanoparticles were also synthesized by coprecipitation, obtaining nanoparticles of two crystallites, 20 and 200 nm. The TEM analysis revealed the presence of larger nanoparticles, displaying low magnetization. Nevertheless, these nanoparticles exhibited a favorable thermal response under the influence of alternating magnetic fields (AMF), with the measured $T_C$ falling within the therapeutic range for magnetic hyperthermia (MH) applications.

The NiCu and NiCr nanoparticles were successfully synthesized using the microemulsion technique. However, in both cases, additional homogenization was required, resulting in partial agglomeration and the formation of larger nanoparticle clusters. Despite achieving a $T_C$ of approximately 45 °C for NiCu nanoparticles, their dissipation and/or heating efficiency (SAR) proved inadequate. We also synthesized NiCu and NiCr nanoparticles of different compositions by mechanical milling. We obtained nanoparticles of different sizes and shapes, which in turn affected their magnetization and thermal response under the influence of AMF, but again the $T_C$ were within the therapeutic temperature range for MH. The sol–gel method proved to be the most promising. We obtained nanoparticles with a size of about 18 nm, which could otherwise be further homogenized. Although the particles partially grew during homogenization, they did not agglomerate because the silica matrix prevented this. The only weakness of silica is that it affects magnetization, but when silica is successfully removed, nanoparticles with higher magnetization and good thermal response under the influence of AMF are obtained. Also, the $T_C$ of the mentioned nanoparticles is in the therapeutic range for use in MH.

Through XRD measurements, we substantiated the successful completion of each synthesis process. This assessment not only validated the creation of magnetic nanoparticles (MNPs) with the intended composition but also allowed for the simultaneous estimation of the MNPs' size, employing Sherrer's equation. The size of the nanoparticles ranged from

5 to 25 nm, and in the case of homogenization, the particles were larger, or the growth of the nanoparticles proved to be a result of the aforementioned homogenization.

The TM curves and corresponding $T_C$ were determined by thermal demagnetization. The $T_C$ is very important in the field of MH. It is the temperature at which the material passes from the magnetic to the non-magnetic state, which means that it no longer gives off heat or heats up under the influence of an external magnetic field. The therapeutic temperature in MH is between 41 and 46 °C. Most of the synthesized nanoparticles have a $T_C$ in the therapeutic range. For maghemite nanoparticles and magnetic NiCu nanoparticles synthesized by the microemulsion method in a silica matrix, this temperature is about 45 °C, and for magnetic NiCu nanoparticles synthesized by the sol–gel method, the $T_C$ varies depending on the composition and ranged from 51 to 64 °C; similarly, for NiCu and NiCr nanoparticles synthesized by mechanical milling, in the case of the former $T_C$ the temperature varies from 24 to 174 °C, while for NiCr nanoparticles it ranges from 69 to 340 °C. As the Ni content increases, so does the $T_C$.

Using TEM analysis, we determined the morphology, size, and size distribution of the synthesized nanoparticles. The size of the nanoparticles, which we estimated using Sherrer's equation, was largely confirmed. The morphology of the MNPs is different, they are mostly round nanoparticles, with the best distribution of the size of the nanoparticles in the case of the synthesis of NiCu nanoparticles by sol–gel synthesis; in these nanoparticles, no agglomeration can be detected. For other nanoparticles, homogenization leads to agglomeration and consequently to a larger size of nanoparticles, and the size distribution is also very wide.

The synthesized nanoparticles show magnetic properties, which we have confirmed by magnetic measurements. The highest magnetization is observed for the maghemite nanoparticles, but it decreases somewhat when they are coated with CM-dextran. The magnetization is much lower for the magnetic NiCu nanoparticles, and the magnetization increases with increasing nickel content. The difference is also evident in the sol–gel method of magnetic NiCu nanoparticles, where the magnetization is about 2 emu/g for particles in the silica matrix, while the value increases to 20 emu/g when the silica is leached.

Measurements of the magnetic heating effect were performed in a conventionally built calorimetric system generating an AMF with a nominal field strength of 2 kA/m up to 42 kA/m in the frequency range from 80 kHz to 800 kHz and equipped with a fiber-optic temperature probe to measure the temperature required to determine SAR. We monitored the temperature response of individually synthesized nanoparticles at different magnetic fields. Depending on the strength of the magnetic field, the nanoparticles showed different temperature responses; all of them mostly reached a heating temperature in the therapeutic range for use in MH, but it varied depending on the composition or strength of the magnetic field. In the case of NiCu nanoparticles synthesized by the microemulsion method, we also calculated the SAR values, which increased with increasing the magnetic field and ranged from 4.3 to 41.6 mW/g, while in the case of magnetic NiCu nanoparticles, the values ranged from 0.12 to 0.60 W/g, depending on the composition.

All synthesized MNPs have properties that are necessary for use in MH. A $T_C$, which is in the therapeutic temperature range, is particularly important because it means that heating in this range is automatically interrupted, which in the case of cancer cells means damage or destruction, while healthy cells remain undamaged. The magnetic NiCu nanoparticles synthesized by the sol–gel method were found to be most suitable for use in MH. They are about 10 nm in size and have a very narrow size distribution. The magnetization is very low in the case of silica, but only this can be removed. $T_C$ is close to the therapeutic temperature, which can also be adjusted by simply changing the composition or the nickel content.

In this short review article, we have shown that it is possible to synthesize MNPs of different compositions with properties for use in MH using various synthetic methods. In addition, it will be necessary to think in the direction of biocompatibility tests, or the use of both in vitro and in vivo tests, especially of the synthesized nanoparticles that have shown the best properties for use in the field MH.

**Funding:** This brief review received no external funding.

**Data Availability Statement:** The data presented in this study are openly.

**Conflicts of Interest:** The authors declare no conflict of interest.

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
