# Peer review of "Magnetic Nanoparticles as Mediators for Magnetic Hyperthermia Therapy Applications: A Status Review"

_applsci, doi:10.3390/app13179548_

Round 1

Reviewer 1 Report

Include cobalt ferrite particles in the abstract as well as it is also used in microfluidics.

The title is a bit misleading, as it gives the impression that the paper itself is a review when the authors report their own work in full detail.

Most of the works included in the literature review are over 10 years old and should include the most up to date works on magnetic particle development and usage.  Please Revise the literature review to include the following:

Hewlin et al created cobalt ferrite particles as magnetic particles (2023) for magnetic manipulators in a ferrofluid for rapid cell separation techniques in microfluidics using magnetophoresis:

Hewlin, R.L., Jr.; Edwards, M.; Schultz, C. Design and Development of a Traveling Wave Ferro-Microfluidic Device and System Rig for Potential Magnetophoretic Cell Separation and Sorting in a Water-Based Ferrofluid. Micromachines 202314, 889. https://doi.org/10.3390/mi14040889

Hewlin et al also recently studied (2023) numerically studied the ability of manipulating magnetite particles as potential drug carriers in a circle of willis model:

Hewlin, R.L., Jr.; Tindall, J.M. Computational Assessment of Magnetic Nanoparticle Targeting Efficiency in a Simplified Circle of Willis Arterial Model. Int. J. Mol. Sci. 202324, 2545. https://doi.org/10.3390/ijms24032545

Ciofani, Gianni, et al. “Magnetic Driven Alginate Nanoparticles for Targeted Drug Delivery.” Current Nanoscience, vol. 4, no. 2, 2008, pp. 212–18, https://doi.org/10.2174/157341308784340886.

Zhao Ling-Yun et al 2013 , Magnetic-mediated hyperthermia for cancer treatment: Research progress and clinical trials, Chinese Phys. B 22 108104

Mozhdeh Peiravi, Hossein Eslami, Mojtaba Ansari, Hadi Zare-Zardini, Magnetic hyperthermia: Potentials and limitations, Journal of the Indian Chemical Society, Volume 99, Issue 1, 2022,

100269,

Refer to other recent works (2020-present date) and add to your review as this will strengthen the manuscript.  As written, the literature lacks sufficient recent works as it pertains to magnetic nanoparticles especially as it pertains to medical drug delivery.

Paragraphs are incredibly long.  For writing purposes, make the paragraphs five sentences with a transition sentence.

Enlarge figures to avoid white space.

Bullet point the potential gaps and room for future work in the conclusion.  As written the review is broad and doesn’t point to specifics in terms of potential areas for improvement.

The conclusion rambles on, you need to get to the “so what” and provide the main takeaway and areas for improvement.

minor english edits  are needed

Reviewer 2 Report

In this manuscript, the authors reviewed the development of magnetic nanoparticles for magnetic hyperthermia therapy applications. This review seems to be useful in the related filed. However, the following problems should be addressed before further consideration of publication:

1. The title needs revision to be concise: “magnetic” appeared for three times here.

2. A scheme can be created after the Introduction section to better demonstrate the contents.

3. I am interested in the reason why Fe2O3, Mg1+xFe2-2xTixO4, Ni1-xCux, CrxNi1-x MNPs are induced. How about Fe3O4 and some other typical types?

4. For a comprehensive review, if possible, the mechanism and process of magnetic hyperthermia should be explained. 

5. The Introduction is suggested to come straight to the point, and references should be enriched. The related advances and applications of magnetic nanoparticles should also be added including: 10.1021/acsami.1c16859, 10.1002/EXP.20210009.

6. Some figures can be merged into a composite one considering that there are too many single figures in the manuscript. All the figures need to be revised with consistent layout/size to improve the readability.

7. The authors can add one special section to propose perspectives or challenges for future researches in this field, and a corresponding schematic figure is suggested be created.

Reviewer 3 Report

Dear authors

This manuscript is interesting and well-written. A detailed and vast method has been performed. There are several comments for the improvement of paper.

Specific comments:

1-      A brief description of methods is need.

2-      Please check this paper:

 Review on magnetic nanoparticles for magnetic Nano fluid hyperthermia application

3-      All figures need reference and permission from publisher

4-       A statement about the importance and introduction of magnetic hyperthermia 

5-      There is scarcity of published papers during 2021 and 2023

6-      This papers can be effective

Metallic Nanoparticles: Their Potential Role in Breast Cancer Immunotherapy via Trained Immunity Provocation

 Applications of metallic nanoparticles in the skin cancer treatment

Nickel Nanoparticles: Applications and Antimicrobial Role against Methicillin-Resistant Staphylococcus aureus Infections

Best regards

Moderate editing of English language required

Round 2

Reviewer 1 Report

All comments addressed

Reviewer 2 Report

I have checked all the revisions.

Reviewer 3 Report

It can be accepted

It is acceptable